# DrNAS:
# Dirichlet Neural Architecture Search

**Xiangning Chen**[1]* **Ruochen Wang**[1]* **Minhao Cheng**[1]* **Xiaocheng Tang**[2] **Cho-Jui Hsieh**[1]

[1]Department of Computer Science, UCLA, [2]DiDi AI Labs

{xiangning, chohsieh}@cs.ucla.edu  {ruocwang, mhcheng}@ucla.edu
xiaochengtang@didiglobal.com

## Abstract

This paper proposes a novel differentiable architecture search method by formulating it into a distribution learning problem. We treat the continuously relaxed architecture mixing weight as random variables, modeled by Dirichlet distribution. With recently developed pathwise derivatives, the Dirichlet parameters can be easily optimized with gradient-based optimizer in an end-to-end manner. This formulation improves the generalization ability and induces stochasticity that naturally encourages exploration in the search space. Furthermore, to alleviate the large memory consumption of differentiable NAS, we propose a simple yet effective progressive learning scheme that enables searching directly on large-scale tasks, eliminating the gap between search and evaluation phases. Extensive experiments demonstrate the effectiveness of our method. Specifically, we obtain a test error of 2.46% for CIFAR-10, 23.7% for ImageNet under the mobile setting. On NAS-Bench-201, we also achieve state-of-the-art results on all three datasets and provide insights for the effective design of neural architecture search algorithms.

## 1 Introduction

Recently, Neural Architecture Search (NAS) has attracted lots of attentions for its potential to democratize deep learning. For a practical end-to-end deep learning platform, NAS plays a crucial role in discovering task-specific architecture depending on users' configurations (e.g., dataset, evaluation metric, etc.). Pioneers in this field develop prototypes based on reinforcement learning (Zoph & Le, 2017), evolutionary algorithms (Real et al., 2019) and Bayesian optimization (Liu et al., 2018). These works usually incur large computation overheads, which make them impractical to use. More recent algorithms significantly reduce the search cost including one-shot methods (Pham et al., 2018; Bender et al., 2018), a continuous relaxation of the space (Liu et al., 2019) and network morphisms (Cai et al., 2018). In particular, Liu et al. (2019) proposes a differentiable NAS framework - DARTS, converting the categorical operation selection problem into learning a continuous architecture mixing weight. They formulate a bi-level optimization objective, allowing the architecture search to be efficiently performed by a gradient-based optimizer.

While current differentiable NAS methods achieve encouraging results, they still have shortcomings that hinder their real-world applications. Firstly, several works have cast doubt on the stability and generalization of these differentiable NAS methods (Chen & Hsieh, 2020; Zela et al., 2020a). They discover that directly optimizing the architecture mixing weight is prone to overfitting the validation set and often leads to distorted structures, e.g., searched architectures dominated by parameter-free operations. Secondly, there exist disparities between the search and evaluation phases, where proxy tasks are usually employed during search with smaller datasets or shallower and narrower networks, due to the large memory consumption of differentiable NAS.

In this paper, we propose an effective approach that addresses the aforementioned shortcomings named Dirichlet Neural Architecture Search (DrNAS). Inspired by the fact that directly optimizing the architecture mixing weight is equivalent to performing point estimation (MLE/MAP) from a probabilistic perspective, we formulate the differentiable NAS as a distribution learning problem

---
*Equal Contribution.

instead, which naturally induces stochasticity and encourages exploration. Making use of the probability simplex property of the Dirichlet samples, DrNAS models the architecture mixing weight as random variables sampled from a parameterized Dirichlet distribution. Optimizing the Dirichlet objective can thus be done efficiently in an end-to-end fashion, by employing the pathwise derivative estimators to compute the gradient of the distribution (Martin Jankowiak, 2018). A straightforward optimization, however, turns out to be problematic due to the uncontrolled variance of the Dirichlet, i.e., too much variance leads to training instability and too little variance suffers from insufficient exploration. In light of that, we apply an additional distance regularizer directly on the Dirichlet concentration parameter to strike a balance between the exploration and the exploitation. We further derive a theoretical bound showing that the constrained distributional objective promotes stability and generalization of architecture search by implicitly controlling the Hessian of the validation error.

Furthermore, to enable a direct search on large-scale tasks, we propose a progressive learning scheme, eliminating the gap between the search and evaluation phases. Based on partial channel connection (Xu et al., 2020), we maintain a task-specific super-network of the same depth and number of channels as the evaluation phase throughout searching. To prevent loss of information and instability induced by partial connection, we divide the search phase into multiple stages and progressively increase the channel fraction via network transformation (Chen et al., 2016). Meanwhile, we prune the operation space according to the learnt distribution to maintain the memory efficiency.

We conduct extensive experiments on different datasets and search spaces to demonstrate DrNAS's effectiveness. Based on the DARTS search space (Liu et al., 2019), we achieve an average error rate of 2.46% on CIFAR-10, which ranks top amongst NAS methods. Furthermore, DrNAS achieves superior performance on large-scale tasks such as ImageNet. It obtains a top-1/5 error of 23.7%/7.1%, surpassing the previous state-of-the-art (24.0%/7.3%) under the mobile setting. On NAS-Bench-201 (Dong & Yang, 2020), we also set new state-of-the-art results on all three datasets with low variance. Our code is available at `https://github.com/xiangning-chen/DrNAS`.

## 2 THE PROPOSED APPROACH

In this section, we first briefly review differentiable NAS setups and generalize the formulation to motivate distribution learning. We then layout our proposed DrNAS and describe its optimization in section 2.2. In section 2.3, we provide a generalization result by showing that our method implicitly regularizes the Hessian norm over the architecture parameter. The progressive architecture learning method that enables direct search is then described in section 2.4.

### 2.1 PRELIMINARIES: DIFFERENTIABLE ARCHITECTURE SEARCH

**Cell-Based Search Space** The cell-based search space is constructed by replications of normal and reduction cells (Zoph et al., 2018; Liu et al., 2019). A normal cell keeps the spatial resolution while a reduction cell halves it but doubles the number of channels. Every cell is represented by a DAG with $N$ nodes and $E$ edges, where every node represents a latent representation $\mathbf{x}^i$ and every edge $(i, j)$ is associated with an operations $o^{(i,j)}$ (e.g., *max pooling* or *convolution*) selected from a predefined candidate space $\mathcal{O}$. The output of a node is a summation of all input flows, i.e., $\mathbf{x}^j = \sum_{i<j} o^{(i,j)}(\mathbf{x}^i)$, and a concatenation of intermediate node outputs, i.e., $concat(\mathbf{x}^2, ..., \mathbf{x}^{N-1})$, composes the cell output, where the first two input nodes $\mathbf{x}^0$ and $\mathbf{x}^1$ are fixed to be the outputs of previous two cells.

**Gradient-Based Search via Continuous Relaxation** To enable gradient-based optimization, Liu et al. (2019) apply a continuous relaxation to the discrete space. Concretely, the information passed from node $i$ to node $j$ is computed by a weighted sum of all operations alone the edge, forming a mixed-operation $\hat{o}^{(i,j)}(x) = \sum_{o \in \mathcal{O}} \theta_o^{(i,j)} o(x)$. The operation mixing weight $\theta^{(i,j)}$ is defined over the probability simplex and its magnitude represents the strength of each operation. Therefore, the architecture search can be cast as selecting the operation associated with the highest mixing weight for each edge. To prevent abuse of terminology, we refer to $\theta$ as the architecture/operation mixing weight, and concentration parameter $\beta$ in DrNAS as the architecture parameter throughout the paper.

**Bilevel-Optimization with Simplex Constraints** With continuous relaxation, the network weight $w$ and operation mixing weight $\theta$ can be jointly optimized by solving a constraint bi-level optimization

problem:

$$\min_{\theta} \ \mathcal{L}_{val}(w^*, \theta) \ \text{ s.t. } \ w^* = \arg\min_{w} \ \mathcal{L}_{train}(w, \theta), \quad \sum_{o=1}^{|\mathcal{O}|} \theta_o^{(i,j)} = 1, \ \forall \ (i,j), \ i < j, \quad (1)$$

where the simplex constraint $\sum_{o=1}^{|\mathcal{O}|} \theta_o^{(i,j)} = 1$ can be either solved explicitly via Lagrangian function (Li et al., 2020), or eliminated by substitution method (e.g., $\theta = Softmax(\alpha), \alpha \in \mathcal{R}^{|\mathcal{O}| \times |E|}$) (Liu et al., 2019). In the next section we describe how this generalized formulation motivates our method.

## 2.2 Differentiable Architecture Search as Distribution Learning

**Learning a Distribution over Operation Mixing Weight** Previous differentiable architecture search methods view the operation mixing weight $\theta$ as learnable parameters that can be directly optimized (Liu et al., 2019; Xu et al., 2020; Li et al., 2020). This has been shown to cause $\theta$ to overfit the validation set and thus induce large generalization error (Zela et al., 2020a;b; Chen & Hsieh, 2020). We recognize that this treatment is equivalent to performing point estimation (e.g., MLE/MAP) of $\theta$ in probabilistic view, which is inherently prone to overfitting (Bishop, 2016; Gelman et al., 2004). Furthermore, directly optimizing $\theta$ lacks sufficient exploration in the search space, and thus cause the search algorithm to commit to suboptimal paths in the DAG that converges faster at the beginning but plateaus quickly (Shu et al., 2020).

Based on these insights, we formulate the differentiable architecture search as a distribution learning problem. The operation mixing weight $\theta$ is treated as random variables sampled from a learnable distribution. Formally, let $q(\theta|\beta)$ denote the distribution of $\theta$ parameterized by $\beta$. The bi-level objective is then given by:

$$\min_{\beta} E_{q(\theta|\beta)}\big[\mathcal{L}_{val}(w^*, \theta)\big] + \lambda d(\beta, \hat{\beta}) \ \text{ s.t. } \ w^* = \arg\min_{w} \ \mathcal{L}_{train}(w, \theta). \quad (2)$$

where $d(\cdot, \cdot)$ is a distance function. Since $\theta$ lies on the probability simplex, we select Dirichlet distribution to model its behavior, i.e., $q(\theta|\beta) \sim Dir(\beta)$, where $\beta$ represents the Dirichlet concentration parameter. Dirichlet distribution is a widely used distribution over the probability simplex (Joo et al., 2019; David M. Blei, 2003; Lee et al., 2020; Kessler et al., 2019), and it enjoys nice properties that enables gradient-based training (Martin Jankowiak, 2018).

The concentration parameter $\beta$ controls the sampling behavior of Dirichlet distribution and is crucial in balancing exploration and exploitation during the search phase. Let $\beta_o$ denote the concentration parameter assign to operation $o$. When $\beta_o \ll 1$ for most $o = 1 \sim |\mathcal{O}|$, Dirichlet tends to produce sparse samples with high variance, reducing the training stability; when $\beta_o \gg 1$ for most $o = 1 \sim |\mathcal{O}|$, the samples will be dense with low variance, leading to insufficient exploration. Therefore, we add a penalty term in the objective (2) to regularize the distance between $\beta$ and the anchor $\hat{\beta} = 1$, which corresponds to the symmetric Dirichlet. In section 2.3, we also derive a theoretical bound showing that our formulation additionally promotes stability and generalization of the architecture search by implicitly regularizing the Hessian of validation loss w.r.t. architecture parameters.

**Learning Dirichlet Parameters via Pathwise Derivative Estimator** Optimizing objective (2) with gradient-based methods requires back-propagation through stochastic nodes of Dirichlet samples. The commonly used reparameterization trick does not apply to Dirichlet distribution, therefore we approximate the gradient of Dirichlet samples via pathwise derivative estimators (Martin Jankowiak, 2018)

$$\frac{d\theta_i}{d\beta_j} = -\frac{\frac{\partial F_{Beta}}{\partial \beta_j}(\theta_j|\beta_j, \beta_{tot} - \beta_j)}{f_{Beta}(\theta_j|\beta_j, \beta_{tot} - \beta_j)} \times \big(\frac{\delta_{ij} - \theta_i}{1 - \theta_j}\big) \quad i, j = 1, ..., |\mathcal{O}|, \quad (3)$$

where $F_{Beta}$ and $f_{Beta}$ denote the CDF and PDF of beta distribution respectively, $\delta_{ij}$ is the indicator function, and $\beta_{tot}$ is the sum of concentrations. $F_{Beta}$ is the iregularised incomplete beta function, for which its gradient can be computed by simple numerical approximation. We refer to (Martin Jankowiak, 2018) for the complete derivations.

**Joint Optimization of Model Weight and Architecture Parameter**   With pathwise derivative estimator, the model weight $w$ and concentration $\beta$ can be jointly optimized with gradient descent. Concretely, we draw a sample $\theta \sim Dir(\beta)$ for every forward pass, and the gradients can be obtained easily through backpropagation. Following DARTS (Liu et al., 2019), we approximate $w^*$ in the lower level objective of equation 2 with one step of gradient descent, and run alternative updates between $w^*$ and $\beta$.

**Selecting the Best Architecture**   At the end of the search phase, a learnt distribution of operation mixing weight is obtained. We then select the best operation for each edge by the most likely operation in expectation:

$$o^{(i,j)} = \arg\max_{o \in \mathcal{O}} E_{q(\theta_o^{(i,j)}|\beta^{(i,j)})}\big[\theta_o^{(i,j)}\big]. \tag{4}$$

In the Dirichlet case, the expectation term is simply the Dirichlet mean $\frac{\beta_o^{(i,j)}}{\sum_{o'} \beta_{o'}^{(i,j)}}$. Note that under the distribution learning framework, we are able to sample a wide range of architectures from the learnt distribution. This property alone has many potentials. For example, in practical settings where both accuracy and latency are concerned, the learnt distribution can be used to find architectures under resource restrictions in a post search phase. We leave these extensions to future work.

## 2.3   THE IMPLICIT REGULARIZATION ON HESSIAN

It has been observed that the generalization error of differentiable NAS is highly related to the dominant eigenvalue of the Hessian of validation loss w.r.t. architecture parameter. Several recent works report that the large dominant eigenvalue of $\nabla_\theta^2 \tilde{\mathcal{L}}_{val}(w, \theta)$ in DARTS results in poor generalization performance (Zela et al., 2020a; Chen & Hsieh, 2020). Our objective (2) is the Lagrangian function of the following constraint objective:

$$\min_\beta E_{q(\theta|\beta)}\big[\mathcal{L}_{val}(w^*, \theta)\big] \text{ s.t. } w^* = \arg\min_w \mathcal{L}_{train}(w, \theta) \,,\, d(\beta, \hat{\beta}) \leq \delta, \tag{5}$$

Here we derive an approximated lower bound based on (5), which demonstrates that our method implicitly controls this Hessian matrix.

**Proposition 1** *Let $d(\beta, \hat{\beta}) = \|\beta - \hat{\beta}\|_2 \leq \delta$ and $\hat{\beta} = 1$ in the bi-level formulation (5). Let $\mu$ denote the mean under the Laplacian approximation of Dirichlet. If $\nabla_\mu^2 \tilde{\mathcal{L}}_{val}(w^*, \mu)$ is Positive Semi-definite, the upper-level objective can be approximated bounded by:*

$$E_{q(\theta|\beta)}(\mathcal{L}_{val}(w, \theta)) \gtrsim \tilde{\mathcal{L}}_{val}(w^*, \mu) + \frac{1}{2}\Big(\frac{1}{1+\delta}(1 - \frac{2}{|\mathcal{O}|}) + \frac{1}{|\mathcal{O}|}\frac{1}{1+\delta}\Big)tr\big(\nabla_\mu^2 \tilde{\mathcal{L}}_{val}(w^*, \mu)\big) \tag{6}$$

*with:*

$$\tilde{\mathcal{L}}_{val}(w^*, \mu) = \mathcal{L}_{val}(w^*, Softmax(\mu)), \ \ \mu_o = \log \beta_o - \frac{1}{|\mathcal{O}|}\sum_{o'} \log \beta_{o'}, \ \ o = 1, \ldots, |\mathcal{O}|.$$

This proposition is driven by the Laplacian approximation to the Dirichlet distribution (MacKay, 1998; Akash Srivastava, 2017). The lower bound (6) indicates that minimizing the expected validation loss controls the trace norm of the Hessian matrix. Empirically, we observe that DrNAS always maintains the dominant eigenvalue of Hessian at a low level (Appendix A.4). The detailed proof can be found in Appendix A.1.

## 2.4   PROGRESSIVE ARCHITECTURE LEARNING

The GPU memory consumption of differentiable NAS methods grows linearly with the size of operation candidate space. Therefore, they usually use a easier proxy task such as training with a smaller dataset, or searching with fewer layers and number of channels (Cai et al., 2019). For instance, the architecture search is performed on 8 cells and 16 initial channels in DARTS (Liu et al., 2019). But during evaluation, the network has 20 cells and 36 initial channels. Such gap makes it hard to derive an optimal architecture for the target task (Cai et al., 2019).

PC-DARTS (Xu et al., 2020) proposes a partial channel connection to reduce the memory overheads of differentiable NAS, where they only send a random subset of channels to the mixed-operation while directly bypassing the rest channels in a shortcut. However, their method causes loss of information and makes the selection of operation unstable since the sampled subsets may vary widely across iterations. This drawback is amplified when combining with the proposed method since we learn the architecture distribution from Dirichlet samples, which already injects certain stochasticity. As shown in Table 1, when directly applying partial channel connection with distribution learning, the test accuracy of the searched architecture decreases over 3% and 18% on CIFAR-10 and CIFAR-100 respectively if we send only 1/8 channels to the mixed-operation.

To alleviate such information loss and instability problem while being memory-efficient, we propose a progressive learning scheme which gradually increases the fraction of channels that are forwarded to the mixed-operation and meanwhile prunes the operation space based on the learnt distribution. We split the search process into consecutive stages and construct a task-specific super-network with the same depth and number of channels as the evaluation phase at the initial stage. Then after each stage, we increase the partial channel fraction, which means that the super-network in the next stage will be wider, i.e., have more convolution channels, and in turn preserve more information. This is achieved by enlarging every convolution weight with a random mapping function similar to Net2Net (Chen et al., 2016). The mapping function $g : \{1, 2, \ldots, q\} \rightarrow \{1, 2, \ldots, n\}$ with $q > n$ is defined as

$$g(j) = \left\{ \begin{array}{ll} j & j \leq n \\ \text{random sample from } \{1, 2, \ldots, n\} & j > n \end{array} \right. \tag{7}$$

To widen layer $l$, we replace its convolution weight $\mathbf{W}^{(l)} \in \mathbb{R}^{Out \times In \times H \times W}$ with a new weight $\mathbf{U}^{(l)}$.

$$\mathbf{U}^{(l)}_{o,i,h,w} = \mathbf{W}^{(l)}_{g(o),g(i),h,w}, \tag{8}$$

where $Out, In, H, W$ denote the number of output and input channels, filter height and width respectively. Intuitively, we copy $\mathbf{W}^{(l)}$ directly into $\mathbf{U}^{(l)}$ and fulfill the rest part by choosing randomly as defined in $g$. Unlike Net2Net, we do not divide $\mathbf{U}^{(l)}$ by a replication factor here because the information flow on each edge has the same scale no matter the partial fraction is. After widening the super-network, we reduce the operation space by pruning out less important operations according to the Dirichlet concentration parameter $\beta$ learnt from the previous stage, maintaining a consistent memory consumption. As illustrated in Table 1, the proposed progressive architecture learning scheme effectively discovers high accuracy architectures and retains a low GPU memory overhead.

## 3    DISCUSSIONS AND RELATIONSHIP TO PRIOR WORK

Early methods in NAS usually include a full training and evaluation procedure every iteration as the inner loop to guide the consecutive search (Zoph & Le, 2017; Zoph et al., 2018; Real et al., 2019). Consequently, their computational overheads are beyond acceptance for practical usage, especially on large-scale tasks.

**Differentiable NAS**   Recently, many works are proposed to improve the efficiency of NAS (Pham et al., 2018; Cai et al., 2018; Liu et al., 2019; Bender et al., 2018; Yao et al., 2020b;a; Mei et al., 2020). Amongst them, DARTS (Liu et al., 2019) proposes a differentiable NAS framework, which introduces a continuous architecture parameter that relaxes the discrete search space. Despite being efficient, DARTS only optimizes a single point on the simplex every search epoch, which has no guarantee to generalize well after the discretization during evaluation. So its stability and generalization have been widely challenged (Li & Talwalkar, 2019; Zela et al., 2020a; Chen & Hsieh, 2020; Wang et al., 2021). Following DARTS, SNAS (Xie et al., 2019) and GDAS (Dong & Yang, 2019)

Table 1: Test accuracy of the derived architectures when searching on NAS-Bench-201 with different partial channel fraction, where $1/K$ channels are sent to the mixed-operation.

| CIFAR-10 | | |
| --- | --- | --- |
| $K$ | **Test Accuracy (%)** | **GPU Memory (MB)** |
| 1 | $94.36 \pm 0.00$ | 2437 |
| 2 | $93.49 \pm 0.28$ | 1583 |
| 4 | $92.85 \pm 0.35$ | 1159 |
| 8 | $91.06 \pm 0.00$ | 949 |
| Ours | $94.36 \pm 0.00$ | 949 |
| CIFAR-100 | | |
| $K$ | **Test Accuracy (%)** | **GPU Memory (MB)** |
| 1 | $73.51 \pm 0.00$ | 2439 |
| 2 | $68.48 \pm 0.41$ | 1583 |
| 4 | $66.68 \pm 3.22$ | 1161 |
| 8 | $55.11 \pm 13.78$ | 949 |
| Ours | $73.51 \pm 0.00$ | 949 |

leverage the gumbel-softmax trick to learn the exact architecture parameter. However, their reparameterization is motivated from reinforcement learning perspective, which is an approximation with softmax rather than an architecture distribution. Besides, their methods require tuning of temperature schedule (Yan et al., 2017; Caglar Gulcehre, 2017). GDAS linearly decreases the temperature from 10 to 1 while SNAS anneals it from 1 to 0.03. In comparison, the proposed method can automatically learn the architecture distribution without the requirement of handcrafted scheduling. BayesNAS (Zhou et al., 2019) applies Bayesian Learning in NAS. Specifically, they cast NAS as model compression problem and use Bayes Neural Network as the super-network, which is difficult to optimize and requires oversimplified approximation. While our method considers the stochasticity in architecture mixing weight, as it is directly related to the generalization of differentiable NAS algorithms (Zela et al., 2020a; Chen & Hsieh, 2020).

**Memory overhead**   When dealing with the large memory consumption of differentiable NAS, previous works mainly restrain the number of paths sampled during the search phase. For instance, ProxylessNAS (Cai et al., 2019) employs binary gates and samples two paths every search epoch. PARSEC (Casale et al., 2019) samples discrete architectures according to a categorical distribution to save memory. Similarly, GDAS (Dong & Yang, 2019) and DSNAS (Hu et al., 2020) both enforce a discrete constraint after the gumbel-softmax reparametrization. However, such discretization manifests premature convergence and cause search instability (Zela et al., 2020b; Zhang et al., 2020). Our experiments in section 4.3 also empirically demonstrate this phenomenon. As an alternative, PC-DARTS (Xu et al., 2020) proposes a partial channel connection, where only a portion of channels is sent to the mixed-operation. However, partial connection can cause loss of information as shown in section 2.4 and PC-DARTS searches on a shallower network with less channels, suffering the search and evaluation gap. Our solution, by progressively pruning the operation space and meanwhile widening the network, searches in a task-specific manner and achieves superior accuracy on challenging datasets like ImageNet (+2.8% over BayesNAS, +2.3% over GDAS, +2.3% over PARSEC, +2.0% over DSNAS, +1.2% over ProxylessNAS, and +0.5% over PC-DARTS).

## 4   EXPERIMENTS

In this section, we evaluate our proposed DrNAS on two search spaces: the CNN search space in DARTS (Liu et al., 2019) and NAS-Bench-201 (Dong & Yang, 2020). For DARTS space, we conduct experiments on both CIFAR-10 and ImageNet in section 4.1 and 4.2 respectively. For NAS-Bench-201, we test all 3 supported datasets (CIFAR-10, CIFAR-100, ImageNet-16-120 (Chrabaszcz et al., 2017)) in section 4.3. Furthermore, we empirically study the dynamics of exploration and exploitation throughout the search process in section 4.4.

### 4.1   RESULTS ON CIFAR-10

**Architecture Space**   For both search and evaluation phases, we stack 20 cells to compose the network and set the initial channel number as 36. We place the reduction cells at the 1/3 and 2/3 of the network and each cell consists of $N = 6$ nodes.

**Search Settings**   We equally divide the 50K training images into two parts, one is used for optimizing the network weights by momentum SGD and the other for learning the Dirichlet architecture distribution by an Adam optimizer. Since Dirichlet concentration $\beta$ must be positive, we apply the shifted exponential linear mapping $\beta = \text{ELU}(\eta) + 1$ and optimize over $\eta$ instead. We use $l_2$ norm to constrain the distance between $\eta$ and the anchor $\hat{\eta} = 0$. The $\eta$ is initialized by standard Gaussian with scale 0.001, and $\lambda$ in (2) is set to 0.001. The ablation study in Appendix A.3 reveals the effectiveness of our anchor regularizer, and DrNAS is insensitive to a wide range of $\lambda$. These settings are consistent for all experiments. For progressive architecture learning, the whole search process consists of 2 stages, each with 25 iterations. In the first stage, we set the partial channel parameter $K$ as 6 to fit the super-network into a single GTX 1080Ti GPU with 11GB memory, i.e., only 1/6 features are sampled on each edge. For the second stage, we prune half candidates and meanwhile widen the network twice, i.e., the operation space size reduces from 8 to 4 and $K$ becomes 3.

**Retrain Settings**   The evaluation phase uses the entire 50K training set to train the network from scratch for 600 epochs. The network weight is optimized by an SGD optimizer with a cosine

Table 2: Comparison with state-of-the-art image classifiers on CIFAR-10.

| Architecture | Test Error (%) | Params (M) | Search Cost (GPU days) | Search Method |
|---|---|---|---|---|
| DenseNet-BC (Huang et al., 2017)$^\star$ | 3.46 | 25.6 | - | manual |
| NASNet-A (Zoph et al., 2018) | 2.65 | 3.3 | 2000 | RL |
| AmoebaNet-A (Real et al., 2019) | $3.34 \pm 0.06$ | 3.2 | 3150 | evolution |
| AmoebaNet-B (Real et al., 2019) | $2.55 \pm 0.05$ | 2.8 | 3150 | evolution |
| PNAS (Liu et al., 2018)$^\star$ | $3.41 \pm 0.09$ | 3.2 | 225 | SMBO |
| ENAS (Pham et al., 2018) | 2.89 | 4.6 | 0.5 | RL |
| DARTS (1st) (Liu et al., 2019) | $3.00 \pm 0.14$ | 3.3 | 0.4 | gradient |
| DARTS (2nd) (Liu et al., 2019) | $2.76 \pm 0.09$ | 3.3 | 1.0 | gradient |
| SNAS (moderate) (Xie et al., 2019) | $2.85 \pm 0.02$ | 2.8 | 1.5 | gradient |
| GDAS (Dong & Yang, 2019) | 2.93 | 3.4 | 0.3 | gradient |
| BayesNAS (Zhou et al., 2019) | $2.81 \pm 0.04$ | 3.4 | 0.2 | gradient |
| ProxylessNAS (Cai et al., 2019)$^\dagger$ | 2.08 | 5.7 | 4.0 | gradient |
| PARSEC (Casale et al., 2019) | $2.81 \pm 0.03$ | 3.7 | 1 | gradient |
| P-DARTS (Chen et al., 2019) | 2.50 | 3.4 | 0.3 | gradient |
| PC-DARTS (Xu et al., 2020) | $2.57 \pm 0.07$ | 3.6 | 0.1 | gradient |
| SDARTS-ADV (Chen & Hsieh, 2020) | $2.61 \pm 0.02$ | 3.3 | 1.3 | gradient |
| GAEA + PC-DARTS (Li et al., 2020) | $2.50 \pm 0.06$ | 3.7 | 0.1 | gradient |
| DrNAS (without progressive learning) | $2.54 \pm 0.03$ | 4.0 | $0.4^\ddagger$ | gradient |
| DrNAS | $2.46 \pm 0.03$ | 4.1 | $0.6^\ddagger$ | gradient |

$^\star$ Obtained without cutout augmentation.
$^\dagger$ Obtained on a different space with PyramidNet (Han et al., 2017) as the backbone.
$^\ddagger$ Recorded on a single GTX 1080Ti GPU.

annealing learning rate initialized as 0.025, a momentum of 0.9, and a weight decay of $3 \times 10^{-4}$. To allow a fair comparison with previous work, we also employ cutout regularization with length 16, drop-path (Zoph et al., 2018) with probability 0.3 and an auxiliary tower of weight 0.4.

**Results** Table 2 summarizes the performance of DrNAS compared with other popular NAS methods, and we also visualize the searched cells in Appendix A.2. DrNAS achieves an average test error of 2.46%, ranking top amongst recent NAS results. ProxylessNAS is the only method that achieves lower test error than us, but it searches on a different space with a much longer search time and has larger model size. We also perform experiments to assign proper credit to the two parts of our proposed algorithm, i.e., Dirichlet architecture distribution and progressive learning scheme. When searching on a proxy task with 8 stacked cells and 16 initial channels as the convention (Liu et al., 2019; Xu et al., 2020), we achieve a test error of 2.54% that surpasses most baselines. Our progressive learning algorithm eliminates the gap between the proxy and target tasks, which further reduces the test error. Consequently, both of the two parts contribute a lot to our performance gains.

## 4.2 RESULTS ON IMAGENET

**Architecture Space** The network architecture for ImageNet is slightly different from that for CIFAR-10 in that we stack 14 cells and set the initial channel number as 48. We also first downscale the spatial resolution from $224 \times 224$ to $28 \times 28$ with three convolution layers of stride 2 following previous works (Xu et al., 2020; Chen et al., 2019). The other settings are the same with section 4.1.

**Search Settings** Following PC-DARTS (Xu et al., 2020), we randomly sample 10% and 2.5% images from the 1.3M training set to alternatively learn network weight and Dirichlet architecture distribution by a momentum SGD and an Adam optimizer respectively. We use 8 RTX 2080 Ti GPUs for both search and evaluation, and the setup of progressive pruning is the same with that on CIFAR-10, i.e., 2 stages with operation space size shrinking from 8 to 4, and the partial channel $K$ reduces from 6 to 3.

**Retrain Settings** For architecture evaluation, we train the network for 250 epochs by an SGD optimizer with a momentum of 0.9, a weight decay of $3 \times 10^{-5}$, and a linearly decayed learning rate initialized as 0.5. We also use label smoothing and an auxiliary tower of weight 0.4 during training. The learning rate warm-up is employed for the first 5 epochs following previous works (Chen et al., 2019; Xu et al., 2020).

Table 3: Comparison with state-of-the-art image classifiers on ImageNet in the mobile setting.

| Architecture | Test Error(%) | | Params | Search Cost | Search |
|---|---|---|---|---|---|
| | top-1 | top-5 | (M) | (GPU days) | Method |
| Inception-v1 (Szegedy et al., 2015) | 30.1 | 10.1 | 6.6 | - | manual |
| MobileNet (Howard et al., 2017) | 29.4 | 10.5 | 4.2 | - | manual |
| ShuffleNet $2\times$ (v1) (Zhang et al., 2018) | 26.4 | 10.2 | $\sim 5$ | - | manual |
| ShuffleNet $2\times$ (v2) (Ma et al., 2018) | 25.1 | - | $\sim 5$ | - | manual |
| NASNet-A (Zoph et al., 2018) | 26.0 | 8.4 | 5.3 | 2000 | RL |
| AmoebaNet-C (Real et al., 2019) | 24.3 | 7.6 | 6.4 | 3150 | evolution |
| PNAS (Liu et al., 2018) | 25.8 | 8.1 | 5.1 | 225 | SMBO |
| MnasNet-92 (Tan et al., 2019) | 25.2 | 8.0 | 4.4 | - | RL |
| DARTS (2nd) (Liu et al., 2019) | 26.7 | 8.7 | 4.7 | 1.0 | gradient |
| SNAS (mild) (Xie et al., 2019) | 27.3 | 9.2 | 4.3 | 1.5 | gradient |
| GDAS (Dong & Yang, 2019) | 26.0 | 8.5 | 5.3 | 0.3 | gradient |
| BayesNAS (Zhou et al., 2019) | 26.5 | 8.9 | 3.9 | 0.2 | gradient |
| DSNAS (Hu et al., 2020)[†] | 25.7 | 8.1 | - | - | gradient |
| ProxylessNAS (GPU) (Cai et al., 2019)[†] | 24.9 | 7.5 | 7.1 | 8.3 | gradient |
| PARSEC (Casale et al., 2019) | 26.0 | 8.4 | 5.6 | 1 | gradient |
| P-DARTS (CIFAR-10) (Chen et al., 2019) | 24.4 | 7.4 | 4.9 | 0.3 | gradient |
| P-DARTS (CIFAR-100) (Chen et al., 2019) | 24.7 | 7.5 | 5.1 | 0.3 | gradient |
| PC-DARTS (CIFAR-10) (Xu et al., 2020) | 25.1 | 7.8 | 5.3 | 0.1 | gradient |
| PC-DARTS (ImageNet) (Xu et al., 2020)[†] | 24.2 | 7.3 | 5.3 | 3.8 | gradient |
| GAEA + PC-DARTS (Li et al., 2020)[†] | 24.0 | 7.3 | 5.6 | 3.8 | gradient |
| DrNAS (without progressive learning)[†] | 24.2 | 7.3 | 5.2 | 3.9 | gradient |
| DrNAS[†] | 23.7 | 7.1 | 5.7 | 4.6 | gradient |

[†] The architecture is searched on ImageNet, otherwise it is searched on CIFAR-10 or CIFAR-100.

**Results**   As shown in Table 3, we achieve a top-1/5 test error of 23.7%/7.1%, outperforming all compared baselines and achieving state-of-the-art performance in the ImageNet mobile setting. The searched cells are visualized in Appendix A.2. Similar to section 4.1, we also report the result achieved with 8 cells and 16 initial channels, which is a common setup for the proxy task on ImageNet (Xu et al., 2020). The obtained 24.2% top-1 accuracy is already highly competitive, which demonstrates the effectiveness of the architecture distribution learning on large-scale tasks. Then our progressive learning scheme further increases the top-1/5 accuracy for 0.5%/0.2%. Therefore, learning in a task-specific manner is essential to discover better architectures.

## 4.3   RESULTS ON NAS-BENCH-201

Recently, some researchers doubt that the expert knowledge applied to the evaluation protocol plays an important role in the impressive results achieved by leading NAS methods (Yang et al., 2020; Li & Talwalkar, 2019). So to further verify the effectiveness of DrNAS, we perform experiments on NAS-Bench-201 (Dong & Yang, 2020), where architecture performance can be directly obtained by querying in the database. NAS-Bench-201 provides support for 3 dataset (CIFAR-10, CIFAR-100, ImageNet-16-120 (Chrabaszcz et al., 2017)) and has a unified cell-based search space containing 15,625 architectures. We refer to their paper (Dong & Yang, 2020) for details of the space. Our experiments are performed in a task-specific manner, i.e., the search and evaluation are based on the same dataset. The hyperparameters for all compared methods are set as their default and for DrNAS, we use the same search settings with section 4.1. We run every method 4 independent times with different random seeds and report the mean and standard deviation in Table 4.

As shown, we achieve the best accuracy on all 3 datasets. On CIFAR-100, we even achieve the global optimal. Specifically, DrNAS outperforms DARTS, GDAS, DSNAS, PC-DARTS, and SNAS by 103.8%, 35.9%, 30.4%, 6.4%, and 4.3% on average. We notice that the two methods (GDAS and DSNAS) that enforce a discrete constraint, i.e., only sample a single path every search iteration, perform undesirable especially on CIFAR-100. In comparison, SNAS, employing a similar Gumbel-softmax trick but without the discretization, performs much better. Consequently, a discrete constraint during search can reduce the GPU memory consumption but empirically suffers instability. In comparison, we develop the progressive learning scheme on top of the architecture distribution learning, enjoying both memory efficiency and strong search performance.

Table 4: Comparison with state-of-the-art NAS methods on NAS-Bench-201.

| Method | CIFAR-10 | | CIFAR-100 | | ImageNet-16-120 | |
|---|---|---|---|---|---|---|
| | validation | test | validation | test | validation | test |
| ResNet (He et al., 2016) | 90.83 | 93.97 | 70.42 | 70.86 | 44.53 | 43.63 |
| Random (baseline) | $90.93 \pm 0.36$ | $93.70 \pm 0.36$ | $70.60 \pm 1.37$ | $70.65 \pm 1.38$ | $42.92 \pm 2.00$ | $42.96 \pm 2.15$ |
| RSPS (Li & Talwalkar, 2019) | $84.16 \pm 1.69$ | $87.66 \pm 1.69$ | $45.78 \pm 6.33$ | $46.60 \pm 6.57$ | $31.09 \pm 5.65$ | $30.78 \pm 6.12$ |
| Reinforce (Zoph et al., 2018) | $91.09 \pm 0.37$ | $93.85 \pm 0.37$ | $70.05 \pm 1.67$ | $70.17 \pm 1.61$ | $43.04 \pm 2.18$ | $43.16 \pm 2.28$ |
| ENAS (Pham et al., 2018) | $39.77 \pm 0.00$ | $54.30 \pm 0.00$ | $10.23 \pm 0.12$ | $10.62 \pm 0.27$ | $16.43 \pm 0.00$ | $16.32 \pm 0.00$ |
| DARTS (1st) (Liu et al., 2019) | $39.77 \pm 0.00$ | $54.30 \pm 0.00$ | $38.57 \pm 0.00$ | $38.97 \pm 0.00$ | $18.87 \pm 0.00$ | $18.41 \pm 0.00$ |
| DARTS (2nd) (Liu et al., 2019) | $39.77 \pm 0.00$ | $54.30 \pm 0.00$ | $38.57 \pm 0.00$ | $38.97 \pm 0.00$ | $18.87 \pm 0.00$ | $18.41 \pm 0.00$ |
| GDAS (Dong & Yang, 2019) | $90.01 \pm 0.46$ | $93.23 \pm 0.23$ | $24.05 \pm 8.12$ | $24.20 \pm 8.08$ | $40.66 \pm 0.00$ | $41.02 \pm 0.00$ |
| SNAS (Xie et al., 2019) | $90.10 \pm 1.04$ | $92.77 \pm 0.83$ | $69.69 \pm 2.39$ | $69.34 \pm 1.98$ | $42.84 \pm 1.79$ | $43.16 \pm 2.64$ |
| DSNAS (Hu et al., 2020) | $89.66 \pm 0.29$ | $93.08 \pm 0.13$ | $30.87 \pm 16.40$ | $31.01 \pm 16.38$ | $40.61 \pm 0.09$ | $41.07 \pm 0.09$ |
| PC-DARTS (Xu et al., 2020) | $89.96 \pm 0.15$ | $93.41 \pm 0.30$ | $67.12 \pm 0.39$ | $67.48 \pm 0.89$ | $40.83 \pm 0.08$ | $41.31 \pm 0.22$ |
| DrNAS | $\mathbf{91.55 \pm 0.00}$ | $\mathbf{94.36 \pm 0.00}$ | $\mathbf{73.49 \pm 0.00}$ | $\mathbf{73.51 \pm 0.00}$ | $\mathbf{46.37 \pm 0.00}$ | $\mathbf{46.34 \pm 0.00}$ |
| optimal | 91.61 | 94.37 | 73.49 | 73.51 | 46.77 | 47.31 |

## 4.4 EMPIRICAL STUDY ON EXPLORATION V.S. EXPLOITATION

We further conduct an empirical study on the dynamics of exploration and exploitation in the search phase of DrNAS on NAS-Bench-201. After every search epoch, We sample 100 $\theta$s from the learned Dirichlet distribution and take the $\arg\max$ to obtain 100 discrete architectures. We then plot the range of their accuracy along with the architecture selected by Dirichlet mean (solid line in Figure 1). Note that in our algorithm, we simply derive the architecture according to the Dirichlet mean as described in Section 2.2. As shown in Figure 1, the accuracy range of the sampled architectures starts very wide but narrows gradually during the search phase. It indicates that DrNAS learns to encourage exploration in the search space at the early stages and then gradually reduces it towards the end as the algorithm becomes more and more confident of the current choice. Moreover, the performance of our architectures can consistently match the best performance of the sampled architectures, indicating the effectiveness of DrNAS.

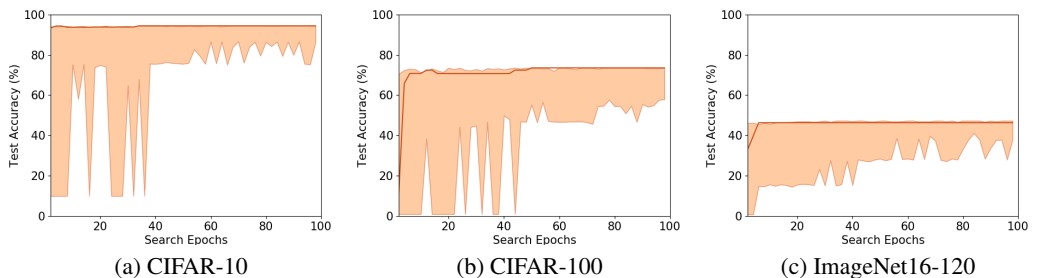

| (a) CIFAR-10 | (b) CIFAR-100 | (c) ImageNet16-120 |

Figure 1: Accuracy range (min-max) of the 100 sampled architectures. Note that the solid line is our derived architecture according to the Dirichlet mean as described in Section 2.2.

## 5 CONCLUSION

In this paper, we propose Dirichlet Neural Architecture Search (DrNAS). We formulate the differentiable NAS as a constraint distribution learning problem, which explicitly models the stochasticity in the architecture mixing weight and balances exploration and exploitation in the search space. The proposed method can be optimized efficiently via gradient-based algorithm, and possesses theoretical benefit to improve the generalization ability. Furthermore, we propose a progressive learning scheme to eliminate the search and evaluation gap. DrNAS consistently achieves strong performance across several image classification tasks, which reveals its potential to play a crucial role in future end-to-end deep learning platform.

## ACKNOWLEDGEMENT

This work is supported in part by NSF under IIS-1901527, IIS-2008173, IIS-2048280 and by Army Research Laboratory under agreement number W911NF-20-2-0158.

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

## A    APPENDIX

### A.1    PROOF OF PROPOSITION 1

**Preliminaries:**    Before the development of Pathwise Derivative Estimator, Laplace Approximate with Softmax basis has been extensively used to approximate the Dirichlet Distribution (MacKay, 1998; Akash Srivastava, 2017). The approximated Dirichlet distribution is:

$$p(\theta(\mathbf{h})|\beta) = \frac{\Gamma(\sum_o \beta_o)}{\prod_o \Gamma(\beta_o)} \prod_o \theta_o^{\beta_o} g(\mathbf{1}^T \mathbf{h}) \tag{9}$$

Where $\theta(\mathbf{h})$ is the softmax-transformed $\mathbf{h}$, $\mathbf{h}$ follows multivariate normal distribution, and $g(\cdot)$ is an arbitrary density to ensure integrability (Akash Srivastava, 2017). The mean $\mu$ and diagonal covariance matrix $\Sigma$ of $\mathbf{h}$ depends on the Dirichlet concentration parameter $\beta$:

$$\mu_o = \log \beta_o - \frac{1}{|\mathcal{O}|} \sum_{o'} \log \beta_{o'} \qquad \Sigma_o = \frac{1}{\beta_o}(1 - \frac{2}{|\mathcal{O}|}) + \frac{1}{|\mathcal{O}|^2} \sum_{o'} \frac{1}{\beta_{o'}} \tag{10}$$

It can be directly obtained from (10) that the Dirichlet mean $\frac{\beta_o}{\sum_{o'} \beta_{o'}} = Softmax(\mu)$. Sampling from the approximated distribution can be down by first sampling from $\mathbf{h}$ and then applying Softmax function to obtain $\theta$. We will leverage the fact that this approximation supports explicit reparameterization to derive our proof.

**Proof:**    Apply the above Laplace Approximation to Dirichlet distribution, the unconstrained upper-level objective in (5) can then be written as:

$$E_{\theta \sim Dir(\beta)} \big[ \mathcal{L}_{val}(w^*, \theta) \big] \tag{11}$$

$$\approx E_{\epsilon \sim \mathcal{N}(0,\Sigma)} \big[ \mathcal{L}_{val}(w^*, Softmax(\mu + \epsilon)) \big] \tag{12}$$

$$\equiv E_{\epsilon \sim \mathcal{N}(0,\Sigma)} \big[ \tilde{\mathcal{L}}_{val}(w^*, \mu + \epsilon) \big] \tag{13}$$

$$\approx E_{\epsilon \sim \mathcal{N}(0,\Sigma)} \big[ \tilde{\mathcal{L}}_{val}(w^*, \mu) + \epsilon^T \nabla_\mu \tilde{\mathcal{L}}_{val}(w^*, \mu) + \frac{1}{2} \epsilon^T \nabla_\mu^2 \tilde{\mathcal{L}}_{val}(w^*, \mu)\epsilon \big] \tag{14}$$

$$= \tilde{\mathcal{L}}_{val}(w^*, \mu) + \frac{1}{2} tr \big( E_{\epsilon \sim \mathcal{N}(0,\Sigma)} \big[ \epsilon \epsilon^T \big] \nabla_\mu^2 \tilde{\mathcal{L}}_{val}(w^*, \mu) \big) \tag{15}$$

$$= \tilde{\mathcal{L}}_{val}(w^*, \mu) + \frac{1}{2} tr \big( \Sigma \nabla_\mu^2 \tilde{\mathcal{L}}_{val}(w^*, \mu) \big) \tag{16}$$

In our full objective, we constrain the Euclidean distance between learnt Dirichlet concentration and fixed prior concentration $||\beta - \mathbf{1}||_2 \le \delta$. The covariance matrix $\Sigma$ of approximated softmax Gaussian can be bounded as:

$$\Sigma_o = \frac{1}{\beta_o}(1 - \frac{2}{|\mathcal{O}|}) + \frac{1}{|\mathcal{O}|^2} \sum_{o'} \frac{1}{\beta_{o'}} \tag{17}$$

$$\ge \frac{1}{1+\delta}(1 - \frac{2}{|\mathcal{O}|}) + \frac{1}{|\mathcal{O}|} \frac{1}{1+\delta} \tag{18}$$

Then (11) becomes:

$$E_{\theta \sim Dir(\beta)} \big[ \mathcal{L}_{val}(w^*, \theta) \big] \tag{19}$$

$$\approx \tilde{\mathcal{L}}_{val}(w^*, \mu) + \frac{1}{2} tr \big( \Sigma \nabla_\mu^2 \tilde{\mathcal{L}}_{val}(w^*, \mu) \big) \tag{20}$$

$$\ge \tilde{\mathcal{L}}_{val}(w^*, \mu) + \frac{1}{2}(\frac{1}{1+\delta}(1 - \frac{2}{|\mathcal{O}|}) + \frac{1}{|\mathcal{O}|} \frac{1}{1+\delta}) tr \big( \nabla_\mu^2 \tilde{\mathcal{L}}_{val}(w^*, \mu) \big) \tag{21}$$

The last line holds when $\nabla_\mu^2 \tilde{\mathcal{L}}_{val}(w^*, \mu)$ is positive semi-definite. In Appendix A.4 we provide an empirical justification for this implicit regularization effect of DrNAS.

### A.2    SEARCHED ARCHITECTURES

We visualize the searched normal and reduction cells in Figure 2 and 3, which is directly searched on CIFAR-10 and ImageNet respectively.

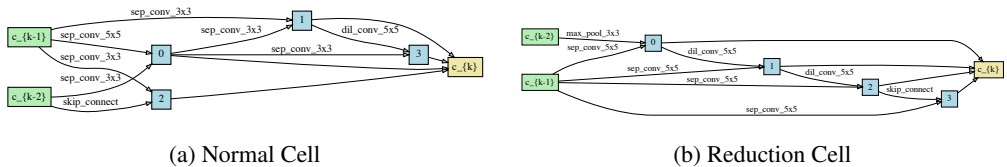

(a) Normal Cell  (b) Reduction Cell

Figure 2: Normal and Reduction cells discovered by DrNAS on CIFAR-10.

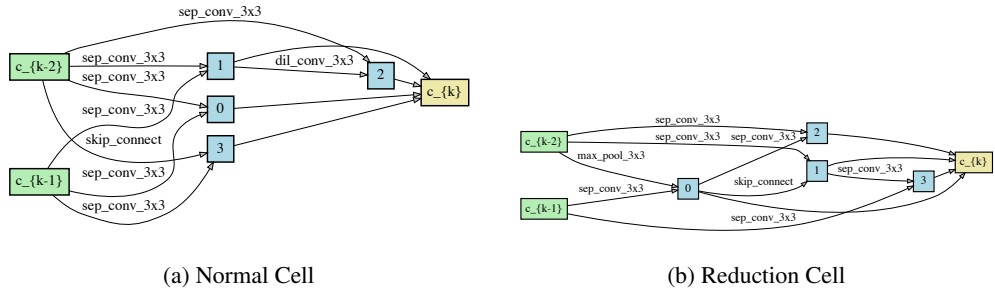

(a) Normal Cell  (b) Reduction Cell

Figure 3: Normal and Reduction cells discovered by DrNAS on ImageNet.

### A.3 ABLATION STUDY ON ANCHOR REGULARIZER PARAMETER $\lambda$

Table 5 shows the accuracy of the searched architecture using different value of $\lambda$ while keeping all other settings the same. Using anchor regularizer? for a wide range of value can boost the accuracy and DrNAS performs quite stable under different $\lambda$s.

Table 5: Test accuracy of the searched architecture with different $\lambda$s on NAS-Bench-201 (CIFAR-10). $\lambda = 1e^{-3}$ is what we used for all of our experiments.

| $\lambda$ | 0 | $5e^{-4}$ | $1e^{-3}$ | $5e^{-3}$ | $1e^{-2}$ | $1e^{-1}$ | 1 |
|---|---|---|---|---|---|---|---|
| Accuracy | 93.78 | 94.01 | 94.36 | 94.36 | 94.36 | 93.76 | 93.76 |

### A.4 EMPIRICAL STUDY ON THE HESSIAN REGULARIZATION EFFECT

We track the anytime Hessian norm on NAS-Bench-201 in Figure 4. The result is obtained by averaging from 4 independent runs. We observe that the largest eigenvalue expands about 10 times when searching by DARTS for 100 epochs. In comparison, DrNAS always maintains the Hessian norm at a low level, which is in agreement with our theoretical analysis in section 2.3. Figure 5 shows the regularization effect under various $\lambda$s. As we can see, DrNAS can keep hessian norm at a low level for a wide range of $\lambda$s, which is in accordance to the relatively stable performance in Table 5.

Moreover, we compare DrNAS with DARTS and R-DARTS on 4 simplified space proposed in (Zela et al., 2020a) and record the endpoint dominant eigenvalue. The first space S1 contains 2 popular operators per edge based on DARTS search result. For S2, S3, and S4, the operation sets are $\{3 \times 3$ *separable convolution*, *skip connection*$\}$, $\{3 \times 3$ *separable convolution*, *skip connection*, *zero*$\}$, and $\{3 \times 3$ *separable convolution*, *noise*$\}$ respectively. As shown in Table 6, DrNAS consistently outperforms DARTS and R-DARTS. The endpoint eigenvalues for DrNAS are 0.0392, 0.0390, 0.0286, 0.0389 respectively. Figure 5 shows the Hessian norm trajectory under various $\lambda$.

### A.5 CONNECTION TO VARIATIONAL INFERENCE

In this section, we draw a connection between DrNAS and Variational Inference (David M. Blei, 2016). We use $w$, $\theta$, and $\beta$ to denote the model weight, operation mixing weight, and Dirichlet concentration parameters respectively, following the main text. The true posterior distribution can

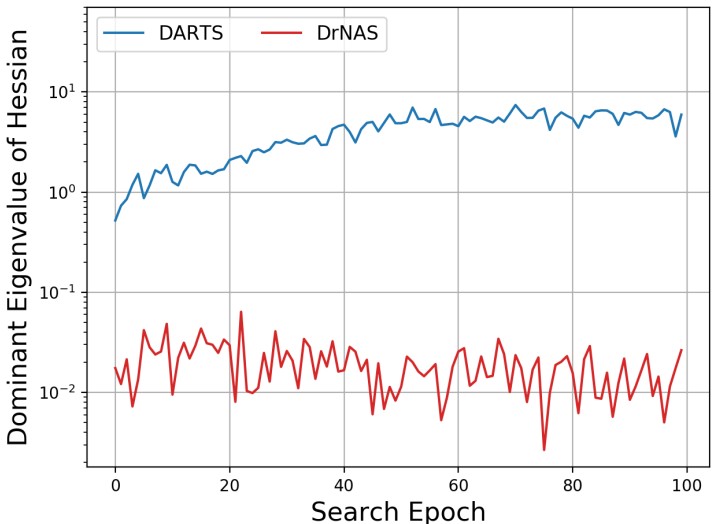

Figure 4: Trajectory of the Hessian norm on NAS-Bench-201 when searching with CIFAR-10 (best viewed in color).

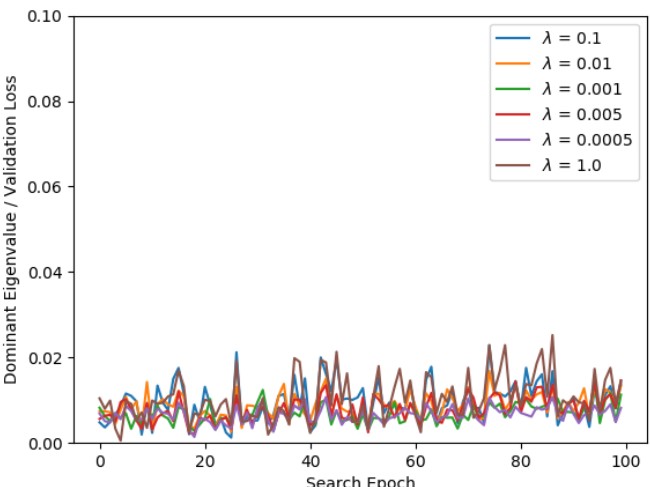

Figure 5: Trajectory of the Hessian norm under various $\lambda$s on NAS-Bench-201 when searching with CIFAR-10 (best viewed in color).

be written as $p(\theta|w, D)$, where $D = \{x_n, y_n\}_{n=1}^{N}$ is the dataset. Let $q(\theta|\beta)$ denote the variational approximation of the true posterior; and assume that $q(\theta|\beta)$ follows Dirichlet distribution. We follow Joo et al. (2019) to assume a symmetric Dirichlet distribution for the prior $p(\theta)$ as well,

Table 6: CIFAR-10 test error on 4 simplified spaces.

|  | s1 | s2 | s3 | s4 |
|---|---|---|---|---|
| DARTS | 3.84 | 4.85 | 3.34 | 7.20 |
| R-DARTS (DP) | 3.11 | 3.48 | 2.93 | 3.58 |
| R-DARTS (L2) | 2.78 | 3.31 | 2.51 | 3.56 |
| DrNAS | **2.74** | **2.47** | **2.4** | **2.59** |

i.e., $p(\theta) = Dir(\mathbf{1})$. The goal is to minimize the KL divergence between the true posterior and the approximated form, i.e., $\min_\beta KL(q(\theta|\beta)||p(\theta|w, D))$. It can be shown that this objective is equivalent to maximizing the evidence lower bound as below (David M. Blei, 2016):

$$\mathcal{L}(\beta) = E_{q(\theta|\beta)}\big[\log p(D|\theta, w)\big] - KL(q(\theta|\beta)||p(\theta|w)) \tag{22}$$

The upper level objective of the bilevel optimization under variational inference framework is then given as:

$$\min_\beta \ E_{q(\theta|\beta)}\big[-\log p(D_{valid}|\theta, w^*)\big] + KL(q(\theta|\beta)||p(\theta)) \tag{23}$$

Note that eq. (23) resembles eq. (2) if we use the negative log likelihood as the loss function and replace $d(\cdot, \cdot)$ with KL divergence. In practice, we find that using a simple l2 distance regularization works well across datasets and search spaces.

