# OpenReview forum: "DrNAS: Dirichlet Neural Architecture Search"
_ICLR.cc/2021/Conference — ICLR 2021 Poster_

### Official Review · AnonReviewer1 · 2020-10-26
**Official Blind Review #1**

**Rating:** 5
**Confidence:** 3

**Review:**

##########################################################################

Summary:

The authors present a new differentiable Neural Architecture Search (NAS) method which places a Dirichlet prior on the edges of a cell for NAS: DrNAS. The authors adapt the original bilevel optimisation to incorporate this new prior over the operation mixing weight. The authors add a regularisation to the Dirichlet parameters and effectively study its effect in an ablation. The authors also introduce a progressive architecture learning to make the optimisation more computationally efficient. DrNAS produces very strong results on three different NAS scenarios and is compared to a wide variety of baselines.


##########################################################################

Reasons for score:

Overall I vote for a weak reject. I think there are some issues surrounding the technical presentation of the work - see the Cons. Also I have some issues with the notation (see below), fixing these will help a lot with the readability of the paper. I have asked for some clarification and I am happy to raise my score in light of clarification from the authors.  The experimental analysis on the other hand is very strong.

##########################################################################

Pros:
Interesting use of the Dirichlet prior for selecting an edge in a cell in the search space for NAS. A hyperparameter is introduced to regularise the Dirichlet parameters and an suitable ablation is provided to demonstrate the regularisation’s sensitivity.
Inference of the prior’s parameters is performed using the new pathwise estimator [4].
Implicitly controlling the largest eigenvalue of the Hessian is very important for producing good results in differentiable NAS [1]. The authors experimentally show that DrNAS is very effective in doing so. I have a question with regards to Proposition 1 (Question 3), if it is indeed correct then the authors also provide an interesting bound to demonstrate that the objective in Equation 2 is implicitly attenuating the largest eigenvalue of the Hessian of the validation loss wrt to the Dirichlet parameters.
DrNAS produces very strong results compared to a variety of benchmarks on three different NAS scenarios.


##########################################################################

Cons:
Claims in the text are not justified in the paper, see Question 2 and Question 3.
General issues surrounding notation**.
The formulation of DrNAS is not strictly formulated within a Bayesian framework with prior and likelihood to infer a posterior over operation mixing weights. The L2 regularization can be formulated as a Gaussian prior about the Dirichlet parameters. Instead the Dirichlet prior is added to the bilevel optimisation in Equation 1 in an ad hoc manner.

##########################################################################

Questions:

Section 2.2, paragraph 1: if \theta is drawn from a distribution why is it less prone to overfitting? The references are text books.
In the paper, for instance in the abstract, you claim that DrNAS encourages better exploration. Of course the model has some stochasticity by sampling from a Dirichlet, but what explicit evidence do you have that having \theta ~ Dirichlet leads to better exploration of the search space for NAS?
In proposition 1: it is assumed that \nabla^2_\mu \tilde{L}_{val} is psd, in general the Hessian of a loss function is pd and psd if L is strictly convex and convex wrt \mu respectively. Does this assumption hold for \tilde{L}_{val}?
You use the pathwise derivative estimator, which is not very popular (I might be wrong here), and you also work the Laplace approximation for Proposition 1 and in related work [3]. What is the advantage of the pathwise derivative estimator versus the Laplace for inference over Dirichlet parameters?
You include stds in Table 4, but not in Table 3? How are the errors in Table 3 calculated?


#########################################################################

Some typos and notations issues and other comments**:

Section 2.2, paragraph 3, first sentence no “the” before “exploration” needed.
Section 2.2, paragraph 3, \beta_0 is not defined.
Equation 2, d(.) not defined.
Equation 3 needs a space.
Title in section 2.3 needs a “the”.
Section 2.3 \alpha and \mu are not defined.
Although these works regard Continual Learning (CL): it is worth mentioning two related works including [5] uses a Dirichlet prior over networks for CL and [6] learns new neurons in a Bayesian NN using an Indian Buffet Process prior.


#########################################################################

[1] A. Zela, T. Elsken, T. Saikia, Y. Marrakchi, T. Brox, and F. Hutter, “Understanding And Robustifying Differentiable Architecture Search,” in ICLR, 2020.
[2] S. Lee, J. Ha, D. Zhang, and G. Kim, “A Neural Dirichlet Process Mixture Model for Task-Free Continual Learning,” in ICLR, 2020.
[3] C. Sutton and A. Srivastava, “Autoencoding Variational Inference for Topic Models,” in ICLR, 2017.
[4] M. Jankowiak and F. Obermeyer, “Pathwise Derivatives Beyond the Reparameterization Trick,” ICML. 2018.
[5] S. Lee, J. Ha, D. Zhang, and G. Kim, “A Neural Dirichlet Process Mixture Model for Task-Free Continual Learning,” in International Conference on Learning Representations, 2020.
[6] S. Kessler, V. Nguyen, S. Zohren, and S. Roberts, “Hierarchical Indian Buffet Neural Networks for Bayesian Continual Learning,” arxiv:1912.02290 2019.

---

> ### Author Response · Authors · 2020-11-19
> **Response to AnonReviewer1**
>
>
> Thank you for your detailed feedback. Below are our responses regarding your questions and concerns.
>
> [Cons]
>
> *a. "The formulation of DrNAS is not strictly formulated within a Bayesian framework with prior and likelihood to infer a posterior over operation mixing weights."*
>
> Thank you for your suggestion on formulating DrNAS in the Bayesian framework. There is a connection between our method and variational inference. We add the connection to Bayesian in Appendix A.5 of the revised paper.
> \
> \
> [Questions]
>
> *a. "if \theta is drawn from a distribution why is it less prone to overfitting? The references are text books."*
>
> This claim is from two lines of research. In the AutoML literature, Nguyen et al. [1] observe that in hyperparameter optimization, the overfitting to the validation set is highly correlated with the curvature of the validation loss. Zela et al. [2] explore it further in the NAS scenario that regularizing the largest dominant eigenvalue can reduce overfitting, as mentioned in our paper. Sampling $\theta$ from Dirichlet implicitly regularizes such Hessian and thus is less prone to overfitting compared with the point estimate of $\theta$ in DARTS, as shown in Figure 2 of the revised paper. In Bayesian literature, if we view $\theta$ as a parameter in the supernet just like the model weights, then modeling this parameter using the Dirichlet distribution is related to the Bayesian Inference for Neural networks. There are some discussions on the effect of the Bayesian Neural Network in reducing overfitting [3, 4, 5].
> \
> \
> *b. "In the paper, for instance in the abstract, you claim that DrNAS encourages better exploration. Of course the model has some stochasticity by sampling from a Dirichlet, but what explicit evidence do you have that having \theta ~ Dirichlet leads to better exploration of the search space for NAS?"*
>
> Thank you for the suggestion. We add a figure (Figure 1) in Section 4.4 of the revised paper to demonstrate it. As we can see, initially the Dirichlet explores more diverse architectures in the space. The exploration is reduced gradually towards the end as DrNAS becomes more confident about the current choice. The shift from exploration to exploitation is controlled by the Dirichlet concentration $\beta$, which is learned from the data.
> \
> \
> *c. "In proposition 1: it is assumed that \nabla^2_\mu \tilde{L}{val} is psd, in general the Hessian of a loss function is pd and psd if L is strictly convex and convex wrt \mu respectively. Does this assumption hold for \tilde{L}{val}?"*
>
> Empirically, we find that the Hessian is close to PSD. For example, on NAS-Bench-201, 92% of its eigenvalues are positive. The eigen spectrum ranges from -0.016 to 0.03. We further study such implicit regularization empirically. As shown in Figure 2 of Appendix A.4, the dominant eigenvalue of Hessian retains at a low level throughout the search phase, whereas its value of DARTS expands about 10 times.
> \
> \
> *d. "You use the pathwise derivative estimator, which is not very popular (I might be wrong here), and you also work the Laplace approximation for Proposition 1 and in related work [3]. What is the advantage of the pathwise derivative estimator versus the Laplace for inference over Dirichlet parameters?"*
>
> Pathwise Derivative for Dirichlet distribution is the de-facto implementation in popular deep learning libraries such as TensorFlow and the latest version of Pytorch. It is very efficient to compute and it is almost as fast as softmax in both forward and backward pass. We use the Laplacian approximation exclusively in our proposition because it facilitates an explicit mathematical form of reparametrization for Dirichlet distribution, which allows us to derive the bound.
> \
> \
> *e. "You include stds in Table 4, but not in Table 3? How are the errors in Table 3 calculated?"*
>
> Most works report a single number on ImageNet due to time complexity, e.g., Table 2 in the PCDARTS paper [6]. We perform another run of the ImageNet experiment and the variance is +- 0.1%.
> \
> \
> [Typos and notations]
>
> Thank you for pointing it out. We fix those issues in the revised draft (colored except for grammatical changes). The citations you recommend are also added to the paper (Section 2.2 paragraph 2).
> \
> \
> [References]
>
> 1. Nyguyen et al. Stable bayesian optimization. JSDSA 2018. doi: 10.1007/s41060-018-0119-9.
> 2. Zela et al. Understanding and Robustifying Differentiable Architecture Search. ICLR 2020
> 3. Blundell et al. Weight Uncertainty in Neural Networks. ICML 2015
> 4. Neal. Bayesian learning for neural networks. 1995
> 5. Mackay. Bayesian interpolation. Neural Computation. 1992
> 6. Xu et al. Partial channel connections for memory-efficient architecture search. ICLR 2020

---

> > ### Author Response · Authors · 2020-11-23
> > **An update on the revised paper**
> >
> > Dear reviewer 1:
> >
> > We add extra figures (Figure 1's bottom row) to show the range of the sampled architectures’ test accuracies (i.e., min-max test accuracies of the 100 sampled architectures). As shown in Figure 1’s bottom row, DrNAS consistently identifies near-optimal architectures according to the Dirichlet mean (as described in Section 2.2).

---

### Official Review · AnonReviewer3 · 2020-10-28
**Official Blind Review #3**

**Rating:** 6
**Confidence:** 3

**Review:**

This paper proposes a differentiable NAS algorithm based on the Dirichlet architecture distribution. Different from the previous differentiable and stochastic NAS algorithms that used the Gumbel-softmax trick or the Categorical distribution, the proposed DrNAS does not require any temperature scheduling for balancing exploration and exploitation, and moreover it does not suffer premature convergence and instability during search. In addition, in order to reduce the memory consumption when searching based on the super-net that mixes all possible operations, the proposed DrNAS applies the progressive learning scheme by combining the network widening with the operation pruning.

Pros.
The proposed algorithm is technically sound, and the motivation and both of the theoretical and empirical analysis are reasonable to support the use of the Dirichlet architecture distribution with the progressive learning. The experimental results also show that the proposed DrNAS consistently outperforms all previous NAS algorithms on CIFAR-10, ImageNet, and especially NAS-Bench-201.

Cons.
My main concern of the proposed method is how to produce the sparse solution during search to reduce the architectural bias between the search and retraining phases. This naturally brings up the question of how and when it automatically changes the exploration to the exploitation during search, like a temperature annealing for the Gumbel-softmax trick. How can the proposed method remove the retraining?

When performing NAS on ImageNet, why did the proposed method use the proxy task with the reduced training set, even though it can retain a low memory overhead like a discrete architecture sampling such as DSNAS? How is the performance variance of the proposed NAS method on ImageNet?

Minor: what if the proposed method searches different cell structures for each layer without the repetition?

---

> ### Author Response · Authors · 2020-11-19
> **Response to AnonReviewer3**
>
>
> Thank you for your positive feedback. Below are our responses regarding your concerns.
>
> [Cons]
>
> *a. "My main concern of the proposed method is how to produce the sparse solution during search to reduce the architectural bias between the search and retraining phases. This naturally brings up the question of how and when it automatically changes the exploration to the exploitation during search, like a temperature annealing for the Gumbel-softmax trick. How can the proposed method remove the retraining?"*
>
> Thank you for bringing it up. As you pointed out, some methods enforce sparsity during the search phrase (i.e., single-path sampling-based methods such as GDAS, SNAS at low temperature, and PARSEC). However, these methods suffer from training instability due to model forgetting [1, 2].  In DrNAS, the sparsity and variance of the sampled $\theta$ are fully controlled by the Dirichlet concentration $\beta$, which is learned from the data. It is able to significantly outperform the above mentioned single-path methods across datasets and search spaces (e.g., we achieve 2.46% average test error on CIFAR10, while the number is 2.93%, 2.85%, and 2.81% for GDAS, SNAS, and PARSEC).
>
> In terms of exploration v.s. exploitation. As shown in Figure 1 of the revised paper, DrNAS learns to encourage exploration at the early stages and then gradually reduce it towards the end as the algorithm becomes more confident of the current choice. This is different from Gumbel-softmax as one has to manually tune the temperature annealing schedule carefully. Annealing the temperature too soon results in early convergence, whereas doing it too late will hurt exploitation. For example, although both employ Gumbel-softmax, GDAS uses a completely different scheduling (10 -> 1) from SNAS (1 -> 0.03).
>
> Following most previous differentiable NAS works like DARTS [5] and PCDARTS [3], the best architecture discovered by DrNAS will be retrained from scratch. Our focus is to identify a high accuracy architecture from the search space, rather than producing an already-to-deploy network. We achieve better results than those methods proposed to remove retraining (ImageNet top-1 test error, DrNAS: 23.7%, SNAS: 27.3%, DSNAS: 25.7%).
> \
> \
> *b. "When performing NAS on ImageNet, why did the proposed method use the proxy task with the reduced training set, even though it can retain a low memory overhead like a discrete architecture sampling such as DSNAS? How is the performance variance of the proposed NAS method on ImageNet?"*
>
> Searching on the full ImageNet is computationally expensive on the commonly used DARTS’ space. Therefore, we follow [3, 4] to sample a subset of ImageNet for the search task. In comparison, DSNAS uses the mobilenet space, which is simpler than the DARTS’ space. Empirically the subset is enough for our algorithm to identify a top-performing architecture on ImageNet. We achieve a 23.7% top-1 error, which is much better than the 25.7% top-1 error of DSNAS.
>
> Most works report a single number on ImageNet due to time complexity, e.g., Table 2 in the PCDARTS paper [3]. We perform another run of the ImageNet experiment and the variance is +- 0.1%.
> \
> \
> [minor]
>
> a. “what if the proposed method searches different cell structures for each layer without the repetition?”
>
> We follow the common settings in the differentiable NAS community to construct the search space by repeating normal and reduction cells. We think extending DrNAS to a search space where each cell has a different structure is plausible, and we are happy to investigate it as a next step.
> \
> \
> [References]
>
> 1. Zhang et al. Overcoming Multi-Model Forgetting in One-Shot NAS with Diversity Maximization. CVPR 2020
> 2. Niu et al. Disturbance-immune Weight Sharing for Neural Architecture Search. Arxiv:2003.13089
> 3. Xu et al. Partial channel connections for memory-efficient architecture search. ICLR 2020
> 4. FBNet: Hardware-Aware Efficient ConvNet Design via Differentiable Neural Architecture Search. CVPR 2019
> 5. Liu et al. DARTS: Differentiable Architecture Search. ICLR 2018

---

### Official Review · AnonReviewer4 · 2020-10-29
**Official Blind review #4**

**Rating:** 7
**Confidence:** 4

**Review:**

Summary:
The paper proposes Dirichlet Neural Architecture Search which formulates Neural architecture search as a distribution architecture search problem. They derive a bound that shows that their formulation implicitly regularizes the Hessian norm with respect to the architecture parameters which has been shown to allow more robust architecture search. They empirically show that their method keeps the dominant eigenvalue of the Hessian much lower compared to DARTS. To reduce memory usage and increase training speed, they incorporate partial channel connections and propose progressively widening the channel connections during training. They demonstrate strong empirical results on several benchmark datasets and NASbench201.

Reasons for score:
Overall I would recommend accepting this paper. It proposes a new efficient distribution learning-based NAS algorithm which regularizes the Hessian with respect to the architecture parameters. The algorithm is demonstrated by efficiently finding high performing networks on Cifar10, Cifar100, and Imagenet. They provide ablation experiments that demonstrate the effects of different parts of their final algorithm. While the benefits of using the Dirichlet distribution could be somewhat better demonstrated empirically separate from the progressive search, compared to existing NAS distribution learning algorithms which don't discretely sample, it is beneficial that the trade-off between exploration and exploitation can be controlled by a penalty term as compared to the popular Gumbel-softmax based methods which arbitrarily anneal the temperature parameter.

Pros:
1. Proposed approach is shown to regularize the Hessian with respect to the architecture parameters which works have shown to lead to better network generalization.

2. The paper provides many results that show that it can find high performing networks quickly with limited compute quite quickly.

3. Provides ablation experiments are which demonstrate the benefits of different parts of the algorithm.

Cons:
1. The paper would be improved greatly if the robustness of the search and final network architectures was empirically explored. This could be accomplished by analyzing the architecture distribution while searching on NASBench-201.

2. The direct effect of the regularizer parameter on the hessian could be better explored. For example something like figure A4 with different values for $\lambda$ would be helpful.

2. The paper would benefit from a proper baseline for the proposed progressive search method. Currently it is somewhat unclear if DrNAS with more time and memory would perform better if used to search the architecture space without a proxy or if the progressive search is regularizing the search to perform better. The reverse experiment of using the progressive search with another distribution learning algorithm would also be beneficial.


Questions:
Was there a particular reason $\lambda$ = 1e-3 was used in the experiments?
In table 4, do you know if there is a particular reason there is no variance in the results for DARTS, ENAS, and your algorithm. It seems a bit strange that DARTS and ENAS all appear to be stuck in the same quite poor performing local minima in this experiment on CIFAR10.


Updates:
Thanks to the authors for addressing my concerns and responding to my questions. The newly added experimental results make the paper stronger and addressed many of my concerns. I recommend this paper to be accepted.

---

> ### Author Response · Authors · 2020-11-19
> **Response to AnonReviewer4 (Part 2: Questions)**
>
> *a. "Was there a particular reason \lambda = 1e-3 was used in the experiments?"*
>
> 1e-3 is commonly used as the scale of the weight decay in differentiable NAS, and we also use this number for our regularization term from the very beginning (without tuning it). The ablation study on this parameter can be found in Appendix A.3. As shown in Table 5, DrNAS’s performance is insensitive to $\lambda$.
> \
> \
> *b. "In table 4, do you know if there is a particular reason there is no variance in the results for DARTS, ENAS, and your algorithm. It seems a bit strange that DARTS and ENAS all appear to be stuck in the same quite poor performing local minima in this experiment on CIFAR10."*
>
> For Table 4, we run the search algorithm 4 times with different random seeds. The final architectures discovered by DrNAS under these 4 independent runs are actually identical. In this case, when we query the 201 database for the architectures' performance, we get the same result, and hence there is no variance. It also shows that the performance of DrNAS is robust to the randomness of the search phase.
>
> ENAS and DARTS also have no variance on NAS-Bench-201. On the 201 space, DARTS always degenerates to a network filled with skip connections when searched on CIFAR10, CIFAR100, and ImageNet16-120. This is the same for ENAS on CIFAR10. Note that in the original NAS-Bench-201 paper [2], ENAS and DARTS also have 0 variance.
> \
> \
> [References]
> 1. Chen et al. Stabilizing Differentiable Architecture Search via Perturbation-based Regularization. ICML 2020
> 2. Dong et al. NAS-Bench-201: Extending the Scope of Reproducible Neural Architecture Search. ICLR 2020
> 3. Xu el al. PC-DARTS: Partial Channel Connections for Memory-Efficient Architecture Search. ICLR 2020

---

> ### Author Response · Authors · 2020-11-19
> **Response to AnonReviewer4 (Part 1: Cons)**
>
>
> Thank you for your positive feedback. Below are our responses regarding your questions and concerns.
>
> *a. "The paper would be improved greatly if the robustness of the search and final network architectures was empirically explored. This could be accomplished by analyzing the architecture distribution while searching on NASBench-201."*
>
> Thank you for your suggestion. Note that for our experiments on NAS-Bench-201 and DARTS’ space, we run both search and retrain phases under 4 random seeds and report the average test error and standard deviation of the derived architecture. As shown in Table 2 and Table 4, the variance of our method is quite low (0% on NAS-Bench-201 and 0.03% on CIFAR10 within DARTS’ space). Specifically, on NAS-Bench-201, the architectures found in 4 independent runs are identical. Moreover, we perform another run of the ImageNet experiment, and the variance is also low (+-0.1%)
>
> Furthermore, we include an experiment in the added Section 4.4 of the revised paper as suggested. Figure 1 in this section examines the architecture distribution while searching on NAS-Bench-201. As shown in Figure 1, DrNAS encourages exploration at the beginning of the training; Towards the end, DrNAS learns to reduce the exploration as the algorithm becomes more confident about its current choice.
> \
> \
> *b. "The direct effect of the regularizer parameter on the hessian could be better explored. For example something like figure A4 with different values for \lambda would be helpful."*
>
> Thank you for your suggestion. We extend Appendix A.4 to discuss this matter. Figure 3 shows the regularization effect under various $\lambda$s. As we can see, DrNAS can keep the hessian norm at a low level for a wide range of $\lambda$s, which is in accordance with the relatively stable performance in 201. Note that the motivation for $\lambda$ is to balance exploration and exploitation.
> \
> \
> *c. "The paper would benefit from a proper baseline for the proposed progressive search method. Currently it is somewhat unclear if DrNAS with more time and memory would perform better if used to search the architecture space without a proxy or if the progressive search is regularizing the search to perform better. The reverse experiment of using the progressive search with another distribution learning algorithm would also be beneficial."*
>
> Thank you for your suggestions. We have an ablation of DrNAS with and without progressive learning on DARTS’ space (Table 2), NAS-Bench-201 (Table 1), and ImageNet (Table 3). Particularly, on NAS-Bench-201, we are able to apply DrNAS without progressive learning on the full search space (a direct search without proxy since this space is quite small). As shown in Table 1, DrNAS with progressive learning (initial K=8, i.e., only 1/8 features are sampled on each edge) achieves the same accuracy as the base DrNAS but consumes less memory. To push the limit of our progressive learning algorithm, we further set the initial K as 16. This setting achieves 93.55% accuracy for CIFAR10 and 69.29% accuracy for CIFAR100 averaged from 4 runs, which are both worse than the base DrNAS (94.36% for CIFAR10 and 73.51% for CIFAR100). So the proposed progressive learning can produce high accuracy networks for a wide range of K (as large as 8 in this example), but its performance can drop if we continue to increase K. Noting that its performance is still better than searching with partial connection but without our progressive learning. Back to your question, the answer is yes, DrNAS (without progressive learning) with more time and memory to search without a proxy can perform better.
>
> Moreover, we ablate the progressive algorithm on the cell-structured space by applying it to DARTS and another sampling-based method SDARTS-RS [1]. With a direct search on the target task, the test error of DARTS on CIFAR10 reduces from 3.0% to 2.72%. We also achieve improvement on SDARTS-RS by reducing its error from 2.67% to 2.64%. All the results are the average over 4 independent runs. We observe more improvement on the original DARTS. This is because that apart from the benefit of a direct search, the partial connection in our progressive learning is shown to have some regularization effect on DARTS [3]. In comparison, the sampling-based method SDARTS-RS and our DrNAS have already regularized the search (regularize the Hessian norm).

---

> > ### Comment · AnonReviewer4 · 2020-11-23
> > **Question about insights from Empirical Study on Exploration vs Exploitation**
> >
> > The results from the Empirical Study on Exploration vs. exploitation seems like it could use some more thorough analysis. It is quite interesting that the algorithm seems to encourage exploration early on. However, that could be related to the stochasticity in picking the most likely architecture early on when the algorithm is converged very far from the uniform initialization. However, the results also seem to indicate that the algorithm does not find the best performing architectures since early on, the variance in the random sampling allows much higher performing architectures to be found. There seems to be a large regret from the final architecture compared to the max validation accuracy architectures found early. Also, there seems to be minimal movement in the average validation accuracy, especially on cifar10

---

> > > ### Author Response · Authors · 2020-11-23
> > > **Response to the follow up question on exploration v.s. exploitation**
> > >
> > > Thank you for your reply. The error bar in Figure 1 (top row) is one standard deviation (± std) of the sampled architectures. So when the variance is very large, the upper edge can be pretty high in the figure. Furthermore, we add extra figures to show the range of the sampled architectures’ test accuracies (i.e., min-max test accuracies of the 100 sampled architectures). As shown in Figure 1’s bottom row, DrNAS consistently identifies near-optimal architectures according to the Dirichlet mean (as described in Section 2.2). Note that the global optimal architecture in the NAS-Bench-201 search space reaches 94.37% test accuracy on CIFAR-10. The final architecture derived by DrNAS achieves 94.36% test accuracy, which is very close to the global optimum. And on CIFAR-100, DrNAS can even achieve the global optimal (73.51% test accuracy) of the whole space. We hope the new figures can resolve your concerns.

---

> > > > ### Author Response · Authors · 2020-11-24
> > > > **Response to the follow up question on exploration v.s. exploitation (cont)**
> > > >
> > > > Thanks again for your reply. We provide a further explanation here. In Figure 1 (top row), we show the accuracy of our derived architecture ± std of the 100 sampled architectures. We choose ± std on top of the derived architecture’s accuracy since it’s a standard way to visualize the variance. When the variance of the sampled architectures is large, the upper edge in the figure can be pretty high. However, there does not exist an architecture that can achieve such high accuracy, which is illustrated by the accuracy range of the sampled architectures (Figure 1’s bottom row) and also the global optimum accuracy in Table 4 (last row). Figure 1’s bottom row also shows that DrNAS performs much better than the random sample. We think the current presentation of Figure 1's top row is a bit confusing and we will replace it with figures only showing the variance in the camera-ready version.

---

### Official Review · AnonReviewer2 · 2020-10-29
**Interesting model; missing insights**

**Rating:** 6
**Confidence:** 2

**Review:**

Summary: This work proposes a modified DARTS optimiser for NAS which assumes a  factorised dirichlet distribution over architecture parameters. It uses pathwise derivatives to learn an MLE estimate of these concentration parameters and adds appropriate regulariser terms to stabilise the training. Furthermore, it employs a protocol for progressively increasing channel fraction to stabilise training within a computation budget. The paper is easy to follow, and relevant experiments have been included.

- With the given probabilistic formulation of this work, it would be useful to include details on how the factorisation of appropriate distributions varies between PARSEC and DrNAS? It might be useful to ground the modelling assumptions within a prior and approximate-posterior framework for better clarity.
- On the same line it would be useful to get insights on how the obtained models qualitatively differ from ProxylassNAS, PARSEC and SNAS.
- The interplay of progressive learning with modelling assumption is a bit unclear. The number of parameters and test error in Table 2 are inversely correlated across SNAS, ProxylessNAS, PARSEC and DrNAS which is perhaps not as surprising. I was wondering if authors have any insights on what aspect of the algo (with and without the two stage progressive learning) contributed to network size.
- Would it be possible to employ the progressive learning policy mentioned in section 4.1 across other DART flavours and understand its impact on model performance?

---

> ### Author Response · Authors · 2020-11-19
> **Response to AnonReviewer2**
>
>
> Thank you for your positive feedback. Below are our responses to your comments.
>
> *a. "With the given probabilistic formulation of this work, it would be useful to include details on how the factorisation of appropriate distributions varies between PARSEC and DrNAS? It might be useful to ground the modelling assumptions within a prior and approximate-posterior framework for better clarity."*
>
> Thank you for your suggestions. As you pointed out, there is a connection between our model with Variational Inference. We add it in Appendix A.5.
>
> DrNAS produces samples of continuous $\theta$ from the Dirichlet distribution, and the continuous $\theta$ will serve as the operation mixing weights directly to train the supernet. The gradient can be backpropagated through the Dirichlet samples efficiently via pathwise derivatives. On the other hand, as mentioned in the related work section, PARSEC samples discrete architectures from a categorical distribution, which is learned by Monte Carlo estimation. The primary goal of its discrete sampling is to reduce the memory cost. However, such discrete sampling has been shown to cause training instability due to model forgetting [1, 2], leading to inferior performance (2.81% test error on CIFAR10, 0.35% worse than DrNAS).
> \
> \
> *b. "On the same line it would be useful to get insights on how the obtained models qualitatively differ from ProxylassNAS, PARSEC and SNAS."*
>
> We focus on comparing the normal cell found by these methods here because normal cells dominate the final architecture (18 out of 20 cells). On DARTS’ search space, the normal cell found by SNAS is a shallow and wide one, and half of the edges are filled with non-parametric skip connections (Figure 2.a of their paper). The lack of depth and parametric operations lead to its comparably poor performance (2.85% on CIFAR10). In contrast, the cell found by DrNAS is deeper and consists of many parametric operations (as shown in Figure 4.a of the revised paper), leading to significantly improved performance (2.46% average test error on CIFAR10). Moreover, the derived architecture of PARSEC (2.81% on CIFAR10) performs worse than ours (2.46% on CIFAR10), possibly due to an excessive number of dilated convolutions (Figure 4 at the bottom of their paper). Most previous works on differentiable NAS use DARTS’ search space. But ProxylessNAS uses its own search space to evaluate the search algorithm, so its obtained model is not quite comparable to ours.
> \
> \
> *c. "The interplay of progressive learning with modelling assumption is a bit unclear. The number of parameters and test error in Table 2 are inversely correlated across SNAS, ProxylessNAS, PARSEC and DrNAS which is perhaps not as surprising. I was wondering if authors have any insights on what aspect of the algo (with and without the two stage progressive learning) contributed to network size."*
>
> Network with more parameters is not guaranteed to produce higher accuracy (e.g., SDARTS-ADV [3] has an error of 2.61% with 3.3M parameters but PARSEC has an error of 2.81% with 3.7M parameters). The focus of our paper is to identify a high accuracy network from the given search space. We propose progressive learning to enable a direct search on the target task, which can narrow the gap between search and evaluation and lead to better performance. As shown in Table 2 and Table 3, the size of the networks discovered by DrNAS with and without progressive learning are roughly the same (4.1M v.s. 4.0M on CIFAR10 and 7.1M v.s. 7.3M on ImageNet), so we think that the model size is primarily related to Dirichlet distribution rather than progressive learning.
> \
> \
> *d. "Would it be possible to employ the progressive learning policy mentioned in section 4.1 across other DART flavors and understand its impact on model performance?"*
>
> Thank you for your suggestions, we ablate the progressive algorithm by applying it to DARTS and another sampling-based method SDARTS-RS [3]. With a direct search on the target task, the test error of DARTS on CIFAR10 reduces from 3.0% to 2.72%. We also achieve improvement on SDARTS-RS by reducing its error from 2.67% to 2.64%. All the results are the average over 4 independent runs. We observe more improvement on the original DARTS. This is because apart from the benefit of a direct search, the partial connection in our progressive learning is shown to have some regularization effect on DARTS [4]. In comparison, the sampling-based method SDARTS-RS and our DrNAS have already regularized the search (regularize the Hessian norm).
> \
> \
> [References]
> 1. Zhang et al. Overcoming Multi-Model Forgetting in One-Shot NAS with Diversity Maximization. CVPR 2020
> 2. Niu et al. Disturbance-immune Weight Sharing for Neural Architecture Search. Arxiv:2003.13089
> 3. Chen et al. Stabilizing Differentiable Architecture Search via Perturbation-based Regularization. ICML 2020
> 4. Xu el al. PC-DARTS: Partial Channel Connections for Memory-Efficient Architecture Search. ICLR 2020

---

### Author Response · Authors · 2020-11-19
**Revision Summary:**

We thank all reviewers for the thoughtful reviews and questions. Our revised paper has been uploaded, with modifications marked red for better visibility. We summarize the changes down below:

* Add an empirical study on exploration and exploitation (Section 4.4) (reviewer 1 and reviewer 4)
* Add a study on the effect of \lambda on Hessian regularization (Appendix A.4) (reviewer 4)
* Add the connection to Bayesian (Appendix A.5) (reviewer 1 and reviewer 2)
* Fix typos and notations (reviewer 1)
* Add the appendix in the pdf submission as well

---

### Decision · Program_Chairs · 2021-01-07
**Final Decision**

**Decision:**

Accept (Poster)

**Comment:**

This paper proposes Dirichlet Neural Architecture Search (DrNAS), a new NAS algorithm that formulates NAS as a distribution architecture search problem. The paper shows theoretically that DrNAS implicitly regularizes the Hessian norm with respect to the architecture parameters (which has been previously shown to allow more robust architecture search) and presents very strong empirical results on several benchmarks, including the tabular benchmark NAS-Bench-201.

The reviews and discussion put this paper very close to the acceptance threshold, so I read it in detail myself to act as a tie-breaker.
I see a lot of positive aspects of this paper:
+ it tackles a very important and timely problem
+ the method implicitly regularizes the Hessian norm with respect to the architecture parameters (which has been previously shown to allow more robust architecture search)
+ empirical results are very strong
+ the paper includes insightful ablation studies for the most important parts of the algorithm
+ the progressive architecture learning approach is a general contribution that reduces the memory complexity and also improves basic DARTS
+ the paper uses a tabular benchmark to yield results that are directly comparable to those of other papers
+ the method's hyperparameters are kept fixed across the different benchmarks, which underlines its robustness.

There are also some negative aspects:
- The method was not originally derived from a Bayesian point of view. Per request of the reviewers, the relation to variational inference has been added to the appendix, but the difference between the L2 regularization and using the explicit KL regularization that falls out of the Bayesian treatment remains. Nevertheless, the new appendix helps to clear up the relationship.

- The paper does not mention availability of the code. This is a must in modern NAS research, as many papers have exposed the poor reproducibility of research in NAS. Fortunately for the authors, I have already seen (independently of this submission) that the code is available on github, but I urge the authors to provide an anonymous repo for review in future submissions, since it was purely by chance that I saw it this time.

- Regarding the point of exploration vs. exploitation, the authors emphasize that in contrast to Gumbel-softmax based methods, such as GDAS and SNAS, with DrNAS there is no need for a cooling schedule. While it is nice to keep the number of hyperparameters small, this also appears to give up control of when the method switches from exploration to exploitation. In practical applications of AutoML, there will be a time budget, and while a cooling schedule can be adapted to fit this budget, it would be suboptimal if DrNAS is still in the exploration phase by the end of the budget, or has already switched to exploitation after, e.g., 5% of the budget. It would be good if the authors could briefly discuss this issue in their final version (if only by acknowledging that this can be a problem).

- Minor negative points
  * The paper sometimes uses jargon, and I believe not even always correctly: e.g., even by googling I did not find such a thing as the "iregularized incomplete beta function", it's also not in the original reference by Jankowiak. I only found the "incomplete beta function" (and the regular one).
  * The author names for several references are garbled. This is likely due to not replacing a comma between the authors with an "and" in the bibtex file.
  * The paper lacks citations for several of the methods it uses, e.g., Adam, cutout, cosine annealing, label smoothing, auxiliary towers, etc. There is no limit on references, and it is standard to cite these concepts to remain more self-contained.
 * The experimental results of GDAS on NB201-CIFAR-100 do not seem to align with the numbers in the NB201 paper. Did you use the numbers from the paper or rerun this method yourself? Please clarify and check this for the final version. The point of tabular benchmarks is to have comparability and consistency across papers!
 * Please have the paper proofread for Grammar again, there are several avoidable errors. E.g., in the first sentence, "lots of attentions" --> "lots of attention". Also things like "alone" -> "along", "down"->"done" etc.

Overall, I think this is a very nice paper, introducing an empirically very strong NAS method that is also theoretically shown to implicitly regularize the Hessian norm with respect to the architecture parameters (which has been previously shown to allow more robust architecture search). I am therefore recommending acceptance. I would like to ask the authors to go through all the reviews again and fix any remaining points in the paper for the final version.